# Evaluation of RANO Criteria for the Assessment of Tumor Progression for Lower-Grade Gliomas

**DOI:** 10.3390/cancers15133274

**Published:** 2023-06-21

**Authors:** Fabio Raman, Alexander Mullen, Matthew Byrd, Sejong Bae, Jinsuh Kim, Houman Sotoudeh, Fanny E. Morón, Hassan M. Fathallah-Shaykh

**Affiliations:** 1Department of Radiology, Johns Hopkins Hospital, 600 N Wolfe St., Baltimore, MD 21287, USA; framan1@jh.edu; 2Department of Radiology, The University of Alabama at Birmingham, Birmingham, AL 35233, USAhsotoudeh@uabmc.edu (H.S.); 3Department of Neurology, The University of Alabama at Birmingham, Birmingham, AL 35233, USA; mjbyrd@uabmc.edu; 4Department of Medicine, O’Neal Comprehensive Cancer Center, The University of Alabama at Birmingham, Birmingham, AL 35233, USA; bsejong@uab.edu; 5Department of Radiology, Emory University, Atlanta, GA 30329, USA; jinsuh.kim@emory.edu; 6Department of Radiology, Baylor College of Medicine, Houston, TX 77030, USA; fmoron@bcm.edu

**Keywords:** Response Assessment in Neuro-Oncology (RANO) criteria, statistical change-of-point method, low-grade gliomas

## Abstract

**Simple Summary:**

Low-grade gliomas (LGGs) are relatively slow-growing primary brain tumors where the clinical criteria for tumor diagnosis and progression assessment include both qualitative and quantitative analytics. The Response Assessment in Neuro-Oncology (RANO) criteria for LGGs define tumor progression as ≥25% change in the T2/FLAIR signal area based on an operator’s discretion of the perpendicular diameter of the largest tumor cross-section. However, sources of error exist, including the limitation of 2D quantification, operator selection of both the tumor cross-section and perpendicular diameters, and the inability to quantify satellite tumor components. The aim of this retrospective study was to assess the accuracy and reproducibility of RANO in LGGs. In a heterogeneous population of 63 participants with different subtypes of LGGs, we showed that the accuracy of RANO compared to visual and volumetric gold standards was, at best, 67% and 57%, respectively. Reproducibility varied widely, even between board-certified neuroradiologists. Our results suggest that advanced approaches, such as computer-assisted tumor segmentation and annotation tools, are necessary to accurately assess LGG progression by reducing human variability.

**Abstract:**

Purpose: The Response Assessment in Neuro-Oncology (RANO) criteria for lower-grade gliomas (LGGs) define tumor progression as ≥25% change in the T2/FLAIR signal area based on an operator’s discretion of the perpendicular diameter of the largest tumor cross-section. Potential sources of error include acquisition inconsistency of 2D slices, operator selection variabilities in both representative tumor cross-section and measurement line locations, and the inability to quantify infiltrative tumor margins and satellite lesions. Our goal was to assess the accuracy and reproducibility of RANO in LG. Materials and Methods: A total of 651 FLAIR MRIs from 63 participants with LGGs were retrospectively analyzed by three blinded attending physicians and three blinded resident trainees using RANO criteria, 2D visual assessment, and computer-assisted 3D volumetric assessment. Results: RANO product measurements had poor-to-moderate inter-operator reproducibility (r^2^ = 0.28–0.82; coefficient of variance (CV) = 44–110%; mean percent difference (diff) = 0.4–46.8%) and moderate-to-excellent intra-operator reproducibility (r^2^ = 0.71–0.88; CV = 31–58%; diff = 0.3–23.9%). When compared to 2D visual ground truth, the accuracy of RANO compared to previous and baseline scans was 66.7% and 65.1%, with an area under the ROC curve (AUC) of 0.67 and 0.66, respectively. When comparing to volumetric ground truth, the accuracy of RANO compared to previous and baseline scans was 21.0% and 56.5%, with an AUC of 0.39 and 0.55, respectively. The median time delay at diagnosis was greater for false negative cases than for false positive cases for the RANO assessment compared to previous (2.05 > 0.50 years, *p* = 0.003) and baseline scans (1.08 > 0.50 years, *p* = 0.02). Conclusion: RANO-based assessment of LGGs has moderate reproducibility and poor accuracy when compared to either visual or volumetric ground truths.

## 1. Introduction

Low-grade gliomas (LGGs) are relatively slow-growing primary brain tumors, designated to World Health Organization (WHO) grade I or II. The clinical criteria for tumor diagnosis and progression assessment include both qualitative and quantitative assessments. Due to the propensity of a delayed anaplastic transformation, a post-operative serial follow-up MRI assessment is warranted in most LGGs. However, an MRI evaluation of LGGs is often complicated by obscure tumor margins and a lack of post-gadolinium enhancement [1], making the visual assessment of tumor progression difficult [2]. Thus, quantitative approaches have started to become incorporated into clinical practice. The Response Assessment in Neuro-Oncology (RANO) criteria for LGGs define tumor progression as ≥ 25% change in the T2/FLAIR signal area based on an operator’s discretion of the perpendicular diameter of the largest tumor cross-section [3]. However, sources of error still exist, including the limitation of 2D quantification, operator selection of both tumor cross-section and perpendicular diameters, and the inability to quantify satellite tumor components.

In recent years, studies have demonstrated that 3D volumetric segmentation provides advantages over 2D tumor assessment that are important to monitor tumor progression and predict patient outcomes [2,4,5,6]. Ellingson et al. demonstrated that the volumetric analysis of Bevacizumab-treated recurrent glioblastoma yields more accurate predictive results than bidirectional measurements based on the RANO criteria [4]. Other studies have shown that these automated methods improve measurement reliability compared to manual assessment, especially in cases with complex tumor heterogeneity [2,5,6]. Specifically, the measurement error increases with tumor size in LGGs, with standard clinical radiological visual reads and 2D quantitative assessments appearing to underestimate tumor expansion [2]. It is important to note that standard clinical workflows largely rely on visual reads by attending radiologists, while 2D RANO assessment is the standard for monitoring tumor progression in clinical trials.

Although studies evaluating various tumor quantification methods have shown promising results, they are limited in sample size, LGG tumor subtypes, and/or cross-analysis of different methods. Our goal was to assess the accuracy and reproducibility of RANO against both visual and volumetric ground truths in a diverse set of LGG tumor subtypes across 63 participants, encompassing 651 FLAIR MRIs.

## 2. Materials and Methods

### 2.1. Study Population and Imaging

All participants (n = 63) were drawn from a previously published retrospective cohort at the University of Alabama at Birmingham (UAB) [7]. All studies were approved by the Institutional Review Board at UAB.

Briefly, consecutive sampling was performed on 165 patients diagnosed with grade 2 gliomas that were seen at UAB clinics from 1 July 2017, to 14 May 2018. A total of 56 gliomas met the inclusion criteria, which included 19 grade 2 oligodendrogliomas, 26 grade 2 astrocytomas, and 11 grade 2 oligoastrocytomas. Only 2 patients received temozolomide (Table 1). All of the oligodendrogliomas had 1p/19q co-deletions except for 1 with a single deletion of 19q. At the time of retrospective review, 34/56 patients had been diagnosed with clinical progression, while the remaining 22/56 were diagnosed as being clinically stable by visual comparison with the most recent MRI. We reviewed the records of 8 patients followed in our clinics for an imaging abnormality without pathological diagnosis; 1 patient was excluded because of lack of follow-up information. All 7 imaging abnormality subjects were considered clinically stable by clinical radiological assessment at the time of review of this study. The imaging abnormality subjects served as negative controls; they were not classified under the other glioma subtypes (oligodendrogliomas, astrocytomas, or oligoastrocytomas).

The inclusion criteria from Fathallah-Shaykh et al. [7] were (1) pathologically diagnosed World Health Organization (WHO) grade 2 oligodendroglioma, grade 2 diffuse astrocytoma, or grade 2 oligoastrocytoma in the supratentorial brain and (2) at least 4 MRI scans available for review, either after the initial diagnosis or after the completion of chemotherapy with temozolomide (if applicable). The LGG MRIs were obtained postoperatively. The exclusion criteria were (1) treatment with radiation therapy after the initial diagnosis or (2) radiological reports indicating development of new enhancement without an increase in FLAIR signal. Patients treated by radiation therapy were excluded because radiation-induced edema may confound assessment of tumor-related FLAIR signal change. We excluded patients whose radiological reports described new enhancing lesions without an increase in FLAIR signal. A total of 651 FLAIR MRIs met the aforementioned criteria and were included for image analysis.

All images were acquired within the clinical diagnostic parameters at the participating institutions, using either a 1.5 or 3.0 Tesla scanner from General Electric (Milwaukee, WI, USA), Philips Medical Systems/Philips Healthcare (Best, The Netherlands), or Siemens (Erlangen, Germany), along with the corresponding head coils. Two-dimensional (2D) Fluid Attenuated Inversion Recovery (FLAIR) images were obtained with the following standard imaging parameters (median, minimum, and maximum values): inversion time (2200, 1400, and 2843 msec), echo time (140, 81, and 395 msec), repetition time (9002, 3900, and 12,000 msec), slice thickness (5, 1, and 6 mm), and interslice gap (6.5 and 0.6 to 7.5 mm).

### 2.2. Image Analysis

RANO measurements and visual assessment were performed by 3 attending physicians and 3 resident trainees blinded to the results of volumetric analysis. RANO product measurements were calculated by multiplying the longest diameter on an axial slice and its longest perpendicular diameter on the same slice. The axial slice was chosen since it was consistently available for all patient scans. Clinical criteria for tumor progression require 25% increase in the product of these two diameters for LGGs. Tumor progression was assessed by comparing the measurement at a given time point to both previous (T − 1) scan and to baseline scan. Positive tumor growth for this study was defined when two out of three attending clinicians’ product measurements were more than 25% larger than previous scan, based on the RANO criteria for tumor progression at a given time point.

For volumetric tumor segmentation, a previously validated method was used that performed automated delineation of the tumor margins, followed by physician review [8]. The volumetric segmentation uses a level set method (LSM) using non-negative matrix factorization (NMF) approach. Briefly, the NMF method is used to discover and identify the image regions—both their homogeneous intensities as well as their spatial distribution. The LSM-NMF method [9] generated 8 segments for every image, which were ranked by their maximal intensities. Second, the final tumor margins were obtained by combining the regions whose maximal intensities were above the level of the gray matter. This tool has been previously validated with the BraTS dataset [10].

Tumor volumes were computed by multiplying the sum of the tumor segments in all axial images by the distance between images, which was performed on the Cheaha supercomputer at UAB. Detection of growth was performed by statistical change-of-point method [7], which was validated by 7 board-certified attending physicians.

For subjective visual assessment, operators performed visual assessments of all 2D slices of the MRIs on whether they believed the tumor displayed growth compared to either the previous scan or to baseline. Positive tumor growth was defined when two out of the three clinicians reached consensus on tumor progression at a given time point. The following diagram provides a schematic of the three different image analysis methods (Figure 1):

### 2.3. Statistical Analyses

All statistical analyses were performed using SPSS Statistics v28.0.1.1 (IBM, Armonk, NY, USA) and Matlab vR2022a (MathWorks, Natick, MA, USA).

Intra- and inter-operator reproducibility was assessed using Pearson correlation, Bland–Altman, and Cohen’s kappa. Univariate regression was used to validate measurements between the two independent operators or sessions, using Pearson’s correlation coefficient (poor agreement = 0; slight = 0.01–0.20; fair = 0.21–0.40; moderate = 0.41–0.60; good = 0.61–0.80; and excellent = 0.81–1.00 agreement). Similar comparisons were conducted using Bland–Altman evaluations to determine limits of agreement and percent differences between pairs of operators (inter-operator) or different runs by one operator (intra-operator). Inter-operator reproducibility was assessed separately across trainees and attendings. Specifically, intra-operator reproducibility was performed by comparing the results of the same operator (attending) performing the analysis 60 days apart. Cohen’s kappa was performed by dichotomizing the subjects into binary variables. Dichotomization was performed with “1” implying growth by either RANO criteria (≥25% change in T2/FLAIR signal) or visual assessment. Statistical significance was defined as *p* < 0.05. Kappa results are interpreted as follows: values ≤ 0 as indicate no agreement, 0.01–0.20 indicate none to slight, 0.21–0.40 indicate fair, 0.41–0.60 indicate moderate, 0.61–0.80 indicate substantial, and 0.81–1.00 indicate almost perfect agreement.

Accuracy assessment was performed by attendings comparing RANO to either visual or volumetric assessment, where 2/3 consensus agreement across operators was required to meet RANO and visual growth criteria for a particular tumor measurement. RANO accuracy assessment was performed compared to both previous and baseline scans. Receiver operating characteristic (ROC) analysis was then used to determine the effectiveness of each RANO method against each of the two ground truths. More specifically, in order to convert the continuous measurements to categorical variables, RANO was assigned binary classifiers, where “0” signifies no growth (<25% change in T2/FLAIR signal) and “1” signifies growth (≥25% change in T2/FLAIR signal) when evaluating the product of perpendicular bidirectional measurements. Binary classification was performed similarly for visual and ground-truth volumetric assessments. For visual assessment, attendings visually inspected tumor growth compared to both previous and baseline scans. For volumetric assessment, the statistical change-of-point method was used to determine the first point of growth [7]. The change-of-point method applies the same rigorous statistical standard to all patients and studies; instead of using the conventional RANO product rule in each dimension universally, the change-of-point statistical method determines if a current measurement is significantly different from all the measurements of the same patient. Logistic regression models were used to examine the accuracy of RANO to distinguish tumor progression based on these binary criteria. Performance of RANO was measured using ROC curves to compare the sensitivity and specificity and to determine the optimum cutoff point. For parameter optimization for the dataset, area under the ROC curve (AUC) was calculated and compared to the value of 0.5 (random agreement) using the methods of Obuchowski et al. [11]. Sensitivity and specificity were reported at the optimum cutoff based on maximum Youden’s index [12]. To evaluate performance between RANO methods comparing baseline and previous scans, ROC curves were compared using the methods of DeLong et al. [13].

Median time delay of diagnosis of tumor progression was assessed between RANO and the volumetric ground truth. False positive was defined as RANO detecting growth prior to volumetric while false negative was defined as RANO detecting growth after volumetric. Both RANO methods compared to previous scan and baseline scan were assessed. Statistical significance was defined as *p* < 0.05.

## 3. Results

### 3.1. RANO Measurements Show Poor-to-Moderate Inter-Operator and Moderate-to-Excellent Intra-Operator Reproducibility

RANO product measurements had moderate inter-operator reproducibility (r^2^ = 0.71–0.82; coefficient of variance (CV) = 81–110%) across the attending operators. In comparison, the inter-operator reproducibility across trainees was poor to moderate for the same measurements (r^2^ = 0.28–0.74; CV = 44–91%, Figure 2). In contrast, systematic bias between operators was greater for attendings (mean percent difference (diff) = 0.4–46.8%) than between trainees (diff = 0.0752–5.95%, Figure 3). RANO had moderate-to-excellent intra-operator reproducibility between the different sets of measurements being performed by the same attending 60 days apart (r^2^ = 0.71–0.88; CV = 31–58%; diff = 0.3–23.9%, Figure 4).

Inter-operator reproducibility assessment using Cohen’s kappa (κ) revealed similar relationships when dichotomizing the presence or absence of growth based on the RANO criteria. Images were assessed by comparing each image to either the subject’s previous or baseline scan (Table 2). The categorical assessment of growth was marginally better between attendings than between trainees (κ_attending_ = 0.172 − 0.532 vs. κ_trainee_ = 0.157 − 0.362), with attendings having slight-to-moderate agreement and trainees having slight-to-fair agreement. Across both groups of raters, comparing images to baseline scan showed a higher reproducibility than comparing to previous scan (κ_baseline_ = 0.275 − 0.532 vs. κ_previous_ = 0.157 − 0.265), with the comparison to the baseline scan showing fair-to-moderate agreement and comparison to previous scan showing slight-to-fair agreement.

For the inter-operator reproducibility between pairs of attendings and trainees, the presence or absence of growth was determined by the RANO criteria (≥25% change in T2/FLAIR signal) between either the previous or baseline scan. Dichotomized measurements were compared. Attendings had slight-to-fair agreement when RANO measurements were assessed compared to the previous scan and fair-to-moderate agreement when compared to the baseline scan. Trainees had slight agreement compared to previous the scan and moderate agreement compared to the baseline scan. Furthermore, since *p* < 0.001 for all cases, the kappa coefficient was significantly different from zero.

### 3.2. RANO Measurements Show Poor Accuracy Compared to Both Visual and Volumetric Ground Truths

When compared to the 3D volumetric ground truth, the accuracy of RANO compared to previous and baseline scans was 21.0% and 56.5% (Table 3), with an AUC of 0.388 and 0.546, respectively (Figure 5A). When compared to the 2D visual ground truth, the accuracy of RANO compared to previous and baseline scans was 66.7% and 65.1% (Table 3), with an AUC of 0.665 and 0.662, respectively (Figure 5B). Visual representations of the example case highlight discordance between RANO and the ground-truth methods of quantification (Figure 6). Although 2D qualitative (visual) and quantitative (RANO) methods may miss tumor growth when focusing on a single slice of the largest tumor cross-section, 3D volumetric assessment is able to detect growth.

The median time delay at diagnosis was greater for false negative cases (i.e., in cases where RANO detected growth after volumetric) than for false positive cases (RANO growth detected earlier growth than volumetric data) for RANO assessment compared to previous scan (2.05 > 0.50 years, *p* = 0.003) and baseline scan (1.08 > 0.50 years, *p* = 0.02, Figure 7). In both situations, the median time delay of >1 year would result in a clinically significant delay in treatment.

The median time delay at diagnosis was greater for false negative cases (i.e., in cases where RANO detected growth after volumetric) than for false positive cases (RANO detected growth earlier than volumetric data) for RANO assessment compared to the previous scan (2.05 > 0.50 years, *p* = 0.003) and baseline scan (1.08 > 0.50 years, *p* = 0.02). In both situations, the median time delay of >1 year would result in a clinically significant delay in treatment.

## 4. Discussion

The RANO criteria used to assess LGGs have slight-to-moderate reproducibility and poor accuracy when comparing to either visual or volumetric ground truths. The RANO criteria are widely utilized to determine therapeutic effectiveness and guide treatment strategies based on the presence or absence of growth in routine clinical practices as well as in clinical trials, mostly for high-grade gliomas [14]. However, the measurements are based on an operator’s discretion of the perpendicular diameter of the largest tumor cross-section. Poor performance of the RANO criteria compared to both visual and volumetric ground truths suggests that alternative methods of evaluating tumor progression in LGGs should be considered for both research and clinical use.

RANO criteria showed slight-to-moderate inter-operator reproducibility when assessing either individual quantitative measurements or their derived dichotomized outcome variables for tumor progression when compared to the percent change from previous or baseline scans. Overall, the RANO criteria were most reproducible across operators when dichotomizing tumor progression based on comparison to the baseline imaging scan. Although in some settings only the previous scan may be available to clinicians, multiple scans are important for alternative methods of measurement, such as the volumetric statistical change-of-point method [7], which was utilized as ground truth in this study. In a clinical setting, having access to a “baseline” scan is helpful to best understand a tumor’s growth trajectory from either the point of first occurrence or a local minimum after undergoing treatment. However, moderate inter-operator reproducibility compared to baseline is unfavorable compared to automated volumetric assessment. Recent evidence suggests that artificial intelligence (AI)-based decision support can improve reproducibility compared to the RANO criteria, with the greatest improvement seen in LGGs (concordance class coefficient = 0.70 for RANO vs. 0.90 with AI) compared to glioblastoma (GBM), suggesting that automated methods are particularly important to reduce variability in LGG assessment [15]. Although the results of our study suggest that intra-operator reproducibility was much higher with moderate-to-excellent agreement, having the same radiologist follow-up on a patient for serial imaging is not practical, especially at large academic centers or private practice groups. Additionally, strong inter-operator reproducibility is essential for standardization and comparing results across clinical trials to evaluate progressive diseases or tumor responses to therapy.

The RANO criteria showed poor accuracy compared to the visual and volumetric ground truths. The volumetric statistical change-of-point method [7] has been previously validated and is considered more accurate than the visual assessment of tumor progression. Compared to the volumetric ground truth, the high false negative rate of up to 48% as well as a median time delay of 1 to 2 years when RANO criteria detect growth after the volumetric assessment could result in a clinically significant delay in treatment, especially as new therapeutic agents are tested in LGG [16,17]. Similarly, the prognostic value of other established tumor classification criteria, such as the Response Evaluation Criteria in Solid Tumors (RECIST) and modified RECIST, was shown to perform inferiorly to volumetric analysis, despite very small inter-operator variability across all three methods [18]. Specifically, these established tumor criteria oversimplify a multidimensional, heterogeneous tumor [19], leading to an underestimation of the extent of tumor responses to therapeutics because of the incorrect quantification of tissue necrosis. Alternatively, automated volumetric assessment tools, such as the methodology described in Kanaly et al. [20], show improved inter-operator reproducibility compared to RECIST and MacDonald criteria for GBM. The limitations of the RANO criteria suggest that novel approaches, such as computer-assisted volumetric tools, are necessary to reduce human variability and accurately assess 3D tumor progression.

Several methods exist that automatically assess the volumes of gliomas and glioblastoma multiforme. The goal of this manuscript was to use a reliable, externally validated methodology to compute the volumetrics. The volumetric tool used to perform the tumor segmentation in this manuscript performed well on the BraTS dataset, the co-winners of the 2016 BraTS competition [8]. The BraTS dataset is a benchmark that researchers use to validate their methodologies in a controlled setting [8,10,21]. The metrics that are important for the potential implementation of volumetric tools in clinical and research settings include accuracy, reproducibility, efficiency, and the ability to predict outcomes. In terms of efficiency, convolutional neural network models require large datasets to effectively train but have performed exceedingly well in terms of dramatically improving efficiency [22,23,24], where Kickingereder et al. also showed how their tool performed well against the RANO criteria in terms of reliability and predicting outcomes [25]. The purpose of this manuscript was to focus on the RANO criteria and evaluate their accuracy and reproducibility, suggesting opportunities for automated algorithms to improve patient outcomes and facilitate research and clinical workflows.

Few other studies have evaluated the reproducibility and accuracy of RANO or other clinical tumor criteria against either visual or volumetric methodologies [2,4,5,6,26,27,28]. More studies evaluate volumetric methodologies against clinical criteria for GBM, likely because LGGs are largely evaluated qualitatively through visual reads in a clinical setting and because the RANO criteria were originally designed for GBM, or high-grade glioma, prior to developing updated guidelines for LGG. However, ongoing and recent clinical trials use the updated RANO criteria for both low- and high-grade gliomas. Gui et al. performed the first-known LGG comparison of clinical radiological criteria, RANO and RECIST, against volumetric segmentation and clinical reads, showing that clinical reads tend to underestimate tumor expansion compared to quantitative techniques. On the other hand, RANO and RECIST tend to either overestimate [27] or underestimate [28] tumor expansion and disease progression, depending on the study. Even manual bidirectional measurements, such as RANO, are limited by the operator’s ability to differentiate between true tumor progression and increased tumor permeability on FLAIR, which is better quantified using a volumetric assessment [4]. Although several studies have shown that automated methods improve measurement reproducibility and accuracy compared to manual assessment [2,5,6,26], the purpose of this study was to perform a comprehensive assessment across six operators using multi-modal statistical analyses in a large cohort of LGGs with a variety of tumor subtypes. The strengths of our study compared to prior studies include the large sample size with a variety of LGG subtypes with long longitudinal follow-up, reproducibility assessment across three attending and three trainees analyzing both the individual RANO product measurements as well as their dichotomized classification based on ≥25% growth criteria, and accuracy assessment against both volumetric and visual ground truths.

Limitations of our study include that fact that the accuracy assessment was only performed for the three attending operators, not the trainees. Conversely, having only board-certified physicians with extensive experience perform the clinical interpretations provides generalizability to the clinical setting where qualitative visual reads are commonly used. Another limitation is that we only focused on LGG, instead of high-grade gliomas where the RANO criteria are more extensively used across standard clinical practice. Further work could evaluate the same question in GBM patients, where the advantages of volumetric methods compared to RANO are not evident [15]. Additionally, additional research should explore whether automated volumetric assessment provides a meaningful clinical benefit, not only higher reproducibility and accuracy. Prospective studies could evaluate whether volumetric-guided patient management improves outcomes for patients compared to the current clinical criteria.

Another limitation of our study is that we did not evaluate sex differences in this manuscript. However, the existing literature suggests that sex differences do exist, with males having higher incidences of both low- and high-grade gliomas with more aggressive subtypes and shorter overall survival times [29,30,31]. Although sex differences impact the outcomes of patients, it is unclear if sex differences affect the ability to characterize tumor progression by RANO. However, the importance of detecting tumor progression earlier in populations with more aggressive phenotypes could have increased clinical significance, where the earlier diagnosis of tumor progression and timely intervention could make a measurable difference in improving outcomes for patients. Furthermore, preliminary evidence suggests that detecting smaller changes in tumor volume growth rates can be effectively used to evaluate the treatment response of LGG for established and investigational agents [16,17]. Huang et al. found that untreated LGG shows a reduction in growth rates after chemotherapy and/or radiotherapy. They suggest that automated volumetric measurements of tumors can serve as a surrogate endpoint for treatment response/disease progression in clinical trials, as this method has higher accuracy and reproducibility [17].

Certain criteria that affect reproducibility include the subjective interpretation of largest cross-sectional slice and longest perpendicular diameter, which is particularly difficult in (1) tumors with complex geometric shapes or (2) heterogeneous tumors with poorly defined margins, predominantly cystic or necrotic lesions, and leptomeningeal tumors. Furthermore, another limitation is the discrepancy between utilizing the largest tumor cross-section or the tumor slice that is most reproducible for measurements [32].

## 5. Conclusions

We showed that the RANO criteria for LGGs have poor accuracy compared to both visual and volumetric assessment as well as moderate-to-high variability across different operators. Although prognosis is better than for GBM, the high false negative rate and extended time delay in detecting progression for LGG compared to the volumetric change-of-point method may lead to a clinically significant delay in treatment. Low variability is important for standardized clinical criteria for guiding response to treatment, monitoring tumor progression in clinical trials, and routine patient management in clinical practice. Automated volumetric assessment methodologies have been shown to improve accuracy, reproducibility, and efficiency for clinicians. The successful deployment of these volumetric tools in clinical practice could potentially facilitate earlier intervention, improved patient outcomes, and reliable monitoring of endpoints in clinical trials.

## Figures and Tables

**Figure 1 cancers-15-03274-f001:**
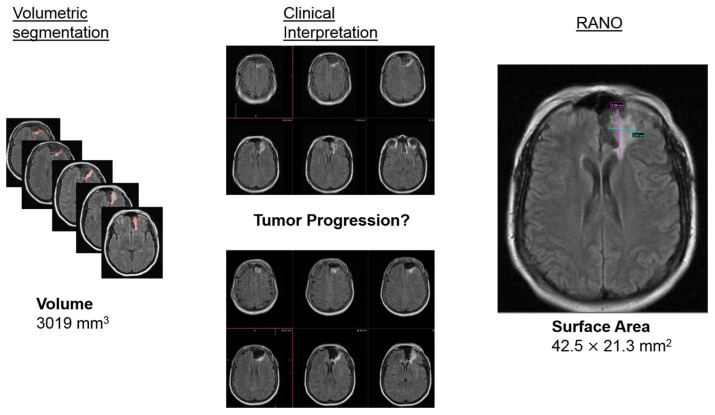
Schematic diagram outlines the 3 different methods for image analysis utilized in the study: Volumetric segmentation (**left**), visual assessment (**middle**), and RANO criteria (**right**).

**Figure 2 cancers-15-03274-f002:**
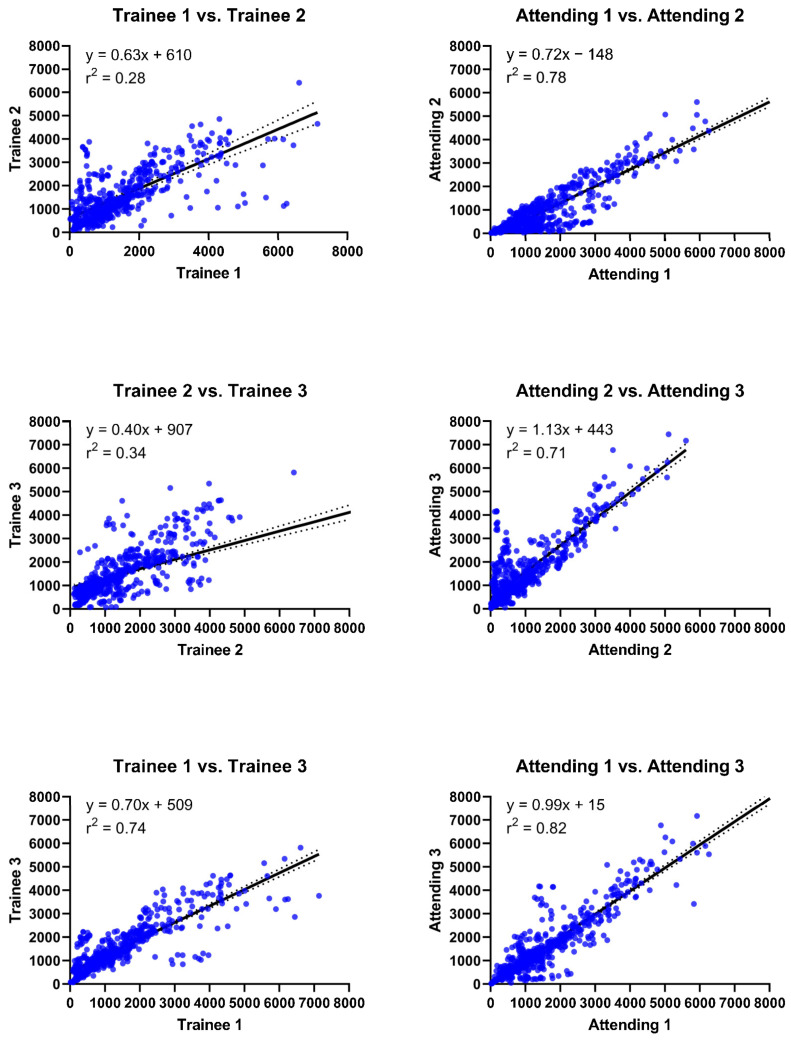
Inter-operator reproducibility assessment using linear regression. Inter-operator comparison of RANO product measurements between each pair of trainees and attendings. Linear regression (solid line) and 95% confidence interval (dotted line) shown with the goodness-of-fit (r^2^) and equation displayed for each bivariate comparison.

**Figure 3 cancers-15-03274-f003:**
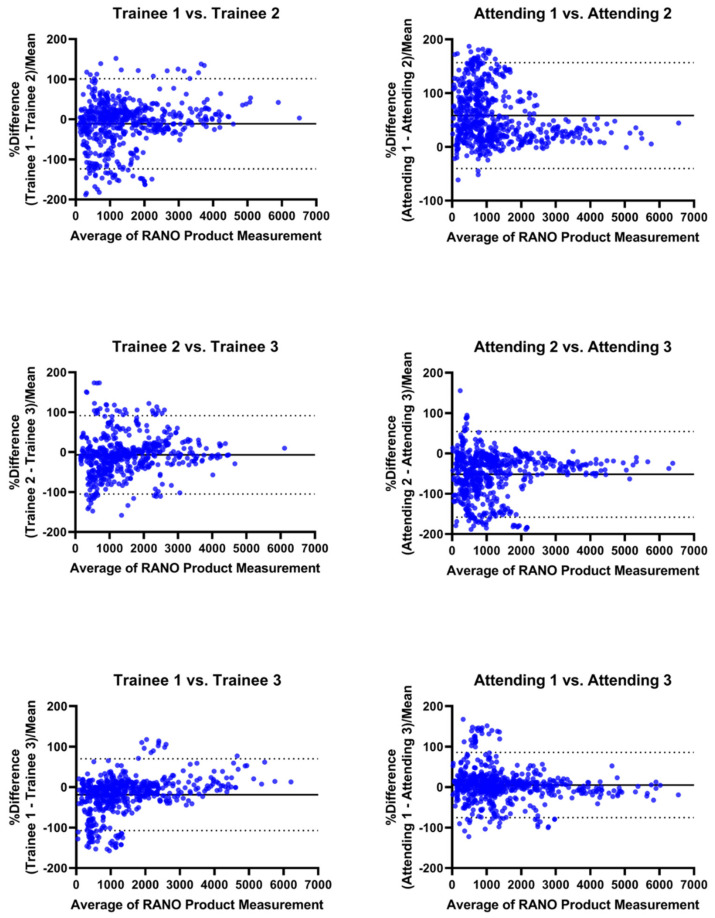
Inter-operator assessment using Bland–Altman. Inter-operator comparison of RANO product measurements between each pair of trainees and attendings. Bland–Altman plots with mean percentile difference (solid line) and 95% confidence intervals (dotted lines). Systematic bias is larger for attendings.

**Figure 4 cancers-15-03274-f004:**
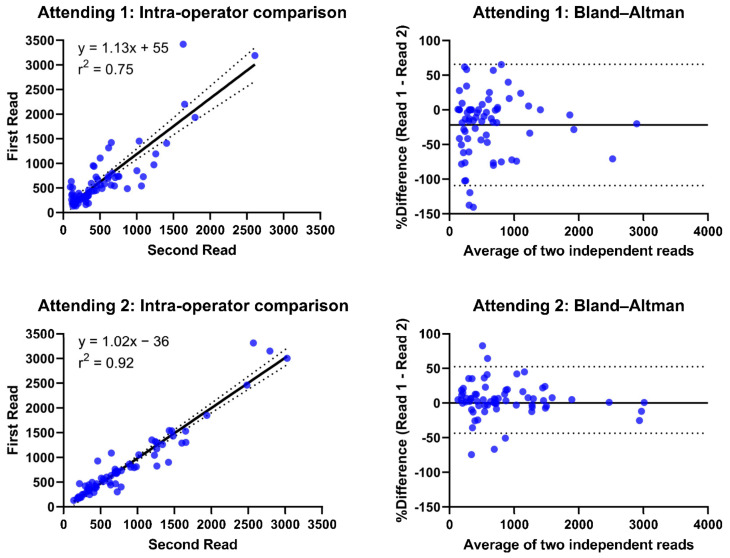
Intra-operator assessment of quantitative RANO product measurements. Intra-operator comparison of RANO product measurements for two attendings evaluating their own measurements against each other performed 1 month apart. Linear regression (solid line) and 95% confidence interval (dotted line) shown with the goodness-of-fit (r^2^) and equation displayed for each intra-operator comparison. Bland–Altman plots with mean difference (solid line) and 95% confidence intervals (dotted lines).

**Figure 5 cancers-15-03274-f005:**
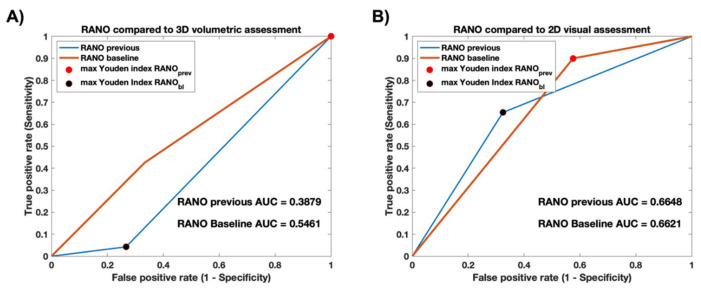
Accuracy assessment compared to 3D volumetric and 2D visual ground truths. Predictor variables: ROC curves for RANO compared to previous and baseline scans. Outcome variables: (**A**) 3D volumetric and (**B**) 2D visual assessment. RANO compared to previous scan has stronger discriminatory ability than RANO compared to baseline scan when evaluating against the 3D volumetric ground truth, but were similar when evaluating against 2D visual assessment. All predictor and outcome variables were dichotomized as “0”, signifying no growth, and “1”, signifying growth. See Section 2 for further details.

**Figure 6 cancers-15-03274-f006:**
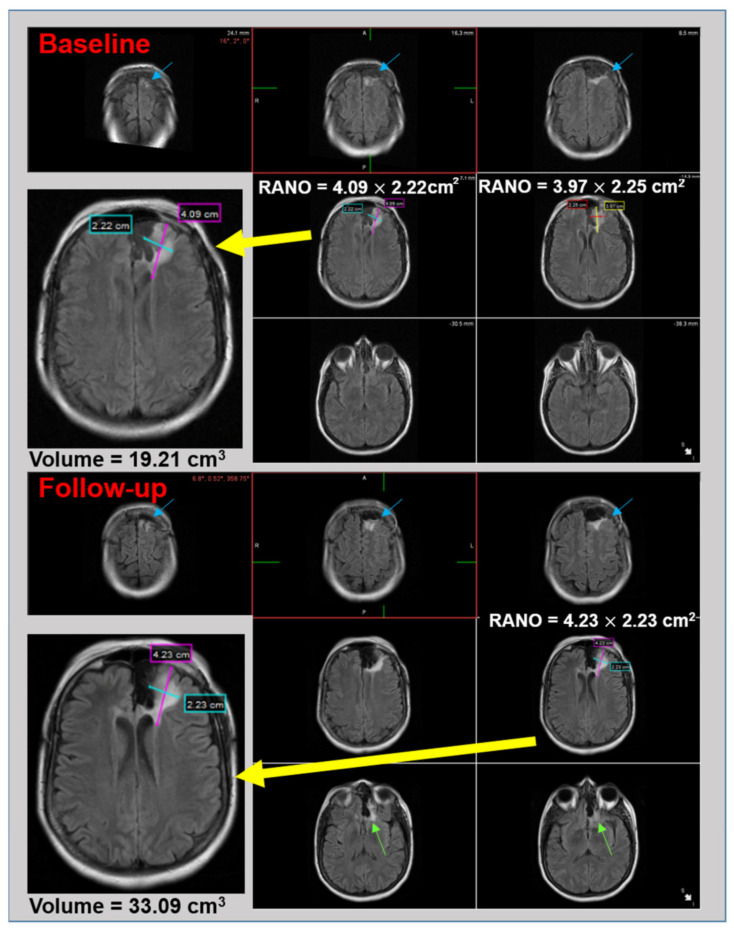
Discrepancy exists between RANO and visual and volumetric ground truths. Representation of subject at baseline (**top**) and after 4-year follow-up (**bottom**). Although 2D RANO shows only a 4% increase in area between baseline (9.08 cm^2^) and follow-up (9.43 cm^2^) and does not meet the criteria for tumor progression, the volume increases by 72% over the same time frame. Visually, the largest 2D tumor cross-sections that are magnified and shown by the yellow arrow are similar between baseline and follow-up. However, 3D volumetric assessment across axial slices reveals tumor growth. The blue arrows highlight sections of the tumor that have grown from baseline to follow-up scan, and the green arrows in the follow-up image show areas of new growth present in the follow-up that were not present in the baseline scan.

**Figure 7 cancers-15-03274-f007:**
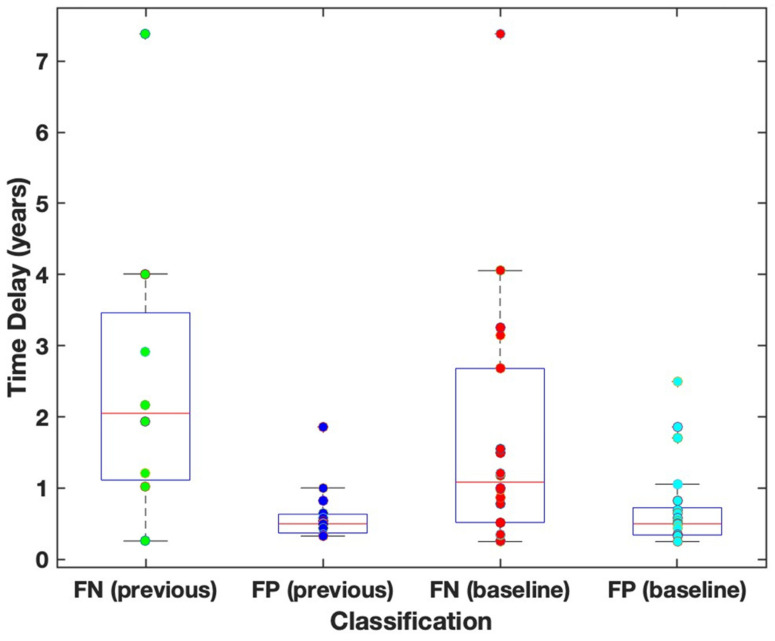
Median time delay in diagnosis of RANO compared to volumetric assessment.

**Table 1 cancers-15-03274-t001:** Subject demographics.

Pathology	Number of Patients	Mean Age (Years)	Number of Males	Number of Females	Number Treated with Temozolomide
Oligodendroglioma	19	47	11	8	1
Astrocytoma	26	46	14	12	1
Mixed glioma	11	53	5	6	0
All	56	48	30	26	2

**Table 2 cancers-15-03274-t002:** Inter-operator reproducibility of growth classification by RANO criteria.

	RANO Compared to Previous Scan Date		RANO Compared to Baseline Scan Date	
	Kappa	*p* Value	Kappa	*p* Value
Attending 1 vs. 2	0.237 ± 0.050	<0.001	0.430 ± 0.037	<0.001
Attending 2 vs. 3	0.265 ± 0.051	<0.001	0.532 ± 0.035	<0.001
Attending 1 vs. 3	0.172 ± 0.049	<0.001	0.275 ± 0.034	<0.001
Trainee 1 vs. 2	0.157 ± 0.054	<0.001	0.362 ± 0.034	<0.001
Trainee 2 vs. 3	0.189 ± 0.049	<0.001	0.332 ± 0.040	<0.001
Trainee 1 vs. 3	0.176 ± 0.049	<0.001	0.275 ± 0.034	<0.001

**Table 3 cancers-15-03274-t003:** Accuracy of RANO compared to 2D visual and 3D volumetric ground truths.

**Ground Truth = 2D Visual Assessment**		
	RANO compared to previous scan date	RANO compared to baseline scan date
**False Negative**	14.29%	4.76%
**False Positive**	19.05%	30.16%
**True Negative**	39.68%	22.22%
**True Positive**	26.98%	42.86%
**Overall accuracy**	66.67%	65.08%
**Ground Truth = 3D Volumetric Assessment**		
**False Negative**	48.39%	9.68%
**False Positive**	30.65%	33.87%
**True Negative**	17.74%	16.13%
**True Positive**	3.23%	40.32%
**Overall accuracy**	20.97%	56.45%

Accuracy of RANO measurement compared to the two reference standards: 2D visual and 3D volumetric assessment. The accuracy rates to the left show RANO compared to previous scan while the column to the far right shows RANO compared to baseline scan. From top to bottom, false negative signifies that the reference standard detected growth prior to RANO; false positive signifies that RANO detected growth prior to the reference standard; true negative signifies that neither method detected growth; and false positive signifies that both methods detected growth at the same time.

## Data Availability

All data are available through links in our group’s *PLoS Medicine* paper, which has been published and is publicly accessible [7].

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
