# Peer review of "Evaluation of RANO Criteria for the Assessment of Tumor Progression for Lower-Grade Gliomas"

_cancers, 2023, doi:10.3390/cancers15133274_

Round 1

Reviewer 1 Report

It is not clear whether imaging follow was obtained after or before surgery in all or in some patients.

"Due to propensity of delayed anaplastic transformation, serial follow-up MRI as-53 assessment is warranted in most LGGs. Please provide reference for this sentence". Most of LGG are now undergoing early surgery in many instances.

"Ellingson et al demonstrated that volumetric anal-65 ysis may help differentiate between true increased tumor expansion and increased tumor 66 permeability on FLAIR, which are difficult to differentiate on bidirectional measurements 67 based on the RANO criteria." What do you mean by tumor permeability on FLAIR?

If I understand well 8 patients didn’t have a pathological diagnosis. This is a potential bias since you cannot be sure that you are dealing with an LGG.

MR parameters are not reported. We don’t know whether volumetric FLAIT were acquired. This might strongly influence RANO evaluation since multiplanar images might not be available.

Details non volumetric evaluation are not reported as well.

Author Response

Response to reviewer #1

Reply to Point 1: Thank you.  The MRIs for the tumor cases were obtained postoperatively (see section 2.1, line 116).

Reply to Point 2: Thank you.  We agree that most of LGG undergo early surgery.  Nonetheless, LGG require post-operative serial MRI assessment (see section 1, line 55). 

Reply to Point 3: Thank you.  We have modified the sentence as follows: Ellingson et al. demonstrated that volumetric analysis of Bevacizumab-treated recurrent glioblastoma yields more accurate predictive results than bidirectional measurements based on the RANO criteria (see section 1, lines 67-69).  The authors hypothesize that the difference may “be explained by the generalized anti-permeability effect of bevacizumab reducing gadolinium diffusion into the interstitium, thereby reducing contrast enhancement, which may be independent of the potential cytostatic effect of bevacizumab” 

Reply to Point 4: Thank you.  These patients served as a negative control (see section 2.1, line 108).

Reply to Point 5: We appreciate the comment regarding the clarification of the MR parameters and the impact of 2D versus 3D FLAIR acquisition. A paragraph containing the relevant parameters has been added to the manuscript (see section 2.1, lines 123-130).

Although we agree that analyzing a 3D T2/FLAIR would have been ideal, our study is a retrospective study in which most of the cases had only axial 2D T2/FLAIR sequences available. The RANO definition for LGG tumor progression was used consistently on the axial plane, as is stated in the manuscript (25% increase of the product of the two-dimensional, perpendicular measurements of the lesion on T2/FLAIR sequence). Lines 135-136 on image analysis has been clarified and reference 3 in manuscript has also been added for further clarification.

Reference

van den Bent, M.J.; Wefel, J.S.; Schiff, D.; Taphoorn, M.J.; Jaeckle, K.; Junck, L.; Armstrong, T.; Choucair, A.; Waldman, A.D.; Gorlia, T.; et al. Response assessment in neuro-oncology (a report of the RANO group): assessment of outcome in trials of diffuse low-grade gliomas. Lancet Oncol 2011, 12, 583-593, doi:10.1016/S1470-2045(11)70057-2.

Reviewer 2 Report

this is a nice paper on evaluation of RANO criteria

I recommend some minor changes:

I would recommend to add specific information on the volumetric segmentation (the citation is not sufficient, since this seems to be a conference proceeding)

the authors could add some information about 3D segmentation tools that are available and compare them, as well as to discuss why they are not used as a standard

there is an error in numbering the rferences (numbers appear in duplicate starting with ref. 10)

Author Response

Responses to reviewer 2

Reply to Point 1: We appreciate the comment regarding additional information on the volumetric segmentation. We’ve added two references to the manuscript (#9 and 10) and addressed this in the paper as follows (see section 2.2, lines 144 – 150):

“The volumetric segmentation uses a level set method (LSM) using non-negative matrix factorization (NMF) approach. Briefly, the NMF method is used to discover and identify the image regions – both their homogeneous intensities as well as their spatial distribution. The LSM-NMF method (Dera, 2016) generated 8 segments for every image which were ranked by their maximal intensities. Second, the final tumor margins were obtained by combining the regions whose maximal intensities were above the level of the gray matter. This tool has been previously validated with the BraTS dataset (Menze, 2015).”

Reply to Point 2: We appreciate the comment on comparing different methods and why alternative methods aren’t used in the manuscript.

Although several 3D segmentation tools exist that perform volumetric assessment of gliomas, the tool that was used in this project was previously validated with the Multimodal Brain Tumor Image Segmentation Benchmark (BRATS) database (Mehta, 2022). The authors of this manuscript competed in the 2016 BRATS challenge and won 1st place by having the most accurate algorithm. The conference proceeding illustrates the work from the 2016 challenge with further details of work outlined in another manuscript (Dera, 2016). However, we agree that several alternative methods could have been used to assess the accuracy of the RANO criteria. The findings of this manuscript simply suggest that alternative quantitative approaches may exist to assessing progression of LGGs. We included a section in discussion comparing different methods (see section 4, lines 387 – 401).

References

Dera, D.; Bouaynaya, N.; Fathallah-Shaykh, H.M. Automated Robust Image Segmentation: Level Set Method Using Nonnegative Matrix Factorization with Application to Brain MRI. Bull Math Biol 2016, 78, 1450-1476, doi:10.1007/s11538-016-0190-0

Menze, B.H.; Jakab, A.; Bauer, S.; Kalpathy-Cramer, J.; Farahani, K.; Kirby, J.; Burren, Y.; Porz, N.; Slotboom, J.; Wiest, R.; et al. The Multimodal Brain Tumor Image Segmentation Benchmark (BRATS). IEEE Trans Med Imaging 2015, 34, 1993-2024, doi:10.1109/TMI.2014.2377694.

Mehta, R.; Filos, A.; Baid, U.; Sako, C.; McKinley, R.; Rebsamen, M.; Datwyler, K.; Meier, R.; Radojewski, P.; Murugesan, G.K.; et al. QU-BraTS: MICCAI BraTS 2020 Challenge on Quantifying Uncertainty in Brain Tumor Segmentation - Analysis of Ranking Scores and Benchmarking Results. J Mach Learn Biomed Imaging 2022, 2022.

Reply to Point 3: Citation numbers have been fixed in the manuscript.

Reviewer 3 Report

The authors present a retrospective analysis of accuracy and reproducibility of RANO to detect tumor growth in low grade gliomas. The manuscript is concise, well-written and will definitely be of interest to the readers of this journal.

Minor:

- the meaning of "AUC" needs to be indicated

- It would be interesting if the authors could develop more on how the 3D automated segmentation and assessment of tumor progression/growth were performed, also including a figure to illustrate the three different methods (RANO vs 2D Visual vs 3D automated).

Author Response

Response to reviewer 3

Reply to Poin1: Area under the ROC curve (AUC) was spelled out in the manuscript at the first occurrence in both the abstract and main body of the paper.

Reply to Point 2: Please see figure 1 that illustrates the different methods (volumetric, visual, and RANO). The diagram has also been added to the method section of the paper. Furthermore, volumetric tool methodology has been further detailed in section 2.2, lines 144-150.

Reviewer 4 Report

The authors report the accuracy and reproducibility evaluation of the Response Assessment in Neuro-Oncology criteria in lower-grade gliomas. Traditional MRI methods for LGGs show limitations in accurately identifying tumor margins and lack post-contrast enhancement. The retrospective study includes 63 patients with 651 FLAIR MRIs regarding inter/intra-operator reproducibility. Results show that the RANO methods for LGGs had poor to moderate inter-operator reproducibility and moderate to excellent intra-operator reproducibility. Therefore, to avoid delays in treatment, automated methods are a vital consideration to reduce variability in LGG assessment. While the current strategy is not optimal for reducing variability, the study provides alternative methods to circumvent the limitations of RANO criteria with a relatively large sample size assessed by 6 operators. I recommend the manuscript for publication in Cancers after the authors address the following issues:

11.  63 participants were included in the study, but in Table 1, only 56 met the inclusion criteria. Can you clarify the other 7? While the authors claimed that “An additional 7 imaging abnormality patients consistent with LGG were included.”

22. Have the authors analyzed the reproducibility and variance difference between male and female patients? Please discuss the key factors contributing to the low or moderate intra/inter-operator reproducibility.

33. Please check references 10 -18. The format and details need to be corrected.

Author Response

Response to reviewer 4

Reply to Point 1: We appreciate the comment regarding the clarification of the 7 additional subjects utilized in the study with imaging abnormalities. Please see details below where we further elaborated in the paper lines 94 – 110.

We reviewed the records of 8 patients followed in our clinics for an imaging abnormality without pathological diagnosis: 1 patient was excluded because of lack of follow-up information. All 7 imaging abnormality patients were considered clinically stable at the time of review of this study. The imaging abnormality subjects served as negative controls; they were not classified under the other glioma subtypes (oligodendrogliomas, astrocytomas, or oligoastrocytomas).

Reply to Point 2: We appreciate the author’s comments regarding potential causes of the low reproducibility and possibility of sex differences in image quantification and interpretation of tumor progression. Please see section 4, lines 442 - 447 and 458 - 465 in the paper.

We did not evaluate sex differences in this manuscript. However, existing literature that sex differences do exist with males having higher incidence of both low- and high-grade gliomas with more aggressive subtypes and shorter overall survival time. Although sex differences impact the outcomes of patients, it’s unclear if the ability to characterize tumor progression by RANO is affected. However, the importance of detecting tumor progression earlier in populations with more aggressive phenotypes could have increased significance clinically where earlier diagnosis of tumor progression and timely intervention could make a measurable difference in improving outcomes for patients.

Certain criteria that affect reproducibility include subjective interpretation of largest cross-sectional slice and longest perpendicular diameter, which is particularly difficult in 1) tumors with complex geometric shapes or 2) heterogeneous tumors with poorly defined margins, predominantly cystic or necrotic lesions, and leptomeningeal tumors. Furthermore, another limitation is the discrepancy between utilizing the largest tumor cross-section or the tumor slice that is most reproducible for measurements.

References

Carrano, A.; Juarez, J.J.; Incontri, D.; Ibarra, A.; Guerrero Cazares, H. Sex-Specific Differences in Glioblastoma. Cells 2021, 10, doi:10.3390/cells10071783.

Sun, T.; Plutynski, A.; Ward, S.; Rubin, J.B. An integrative view on sex differences in brain tumors. Cell Mol Life Sci 2015, 72, 3323-3342, doi:10.1007/s00018-015-1930-2.

Whitmire, P.; Rickertsen, C.R.; Hawkins-Daarud, A.; Carrasco, E., Jr.; Lorence, J.; De Leon, G.; Curtin, L.; Bayless, S.; Clark-Swanson, K.; Peeri, N.C.; et al. Sex-specific impact of patterns of imageable tumor growth on survival of primary glioblastoma patients. BMC Cancer 2020, 20, 447, doi:10.1186/s12885-020-06816-2.

Leao, D.J.; Craig, P.G.; Godoy, L.F.; Leite, C.C.; Policeni, B. Response Assessment in Neuro-Oncology Criteria for Gliomas: Practical Approach Using Conventional and Advanced Techniques. AJNR Am J Neuroradiol 2020, 41, 10-20, doi:10.3174/ajnr.A6358.

Reply to Point 3: The references have been fixed in the manuscript.

Reviewer 5 Report

This is a well designed study that compares subjective visual assessment, RANO measurements and volumetric measurements for LGG. The methodology is sound, and the results are succinctly presented.

Below are some of the drawbacks of the study.

1. Line 113- The RANO measurements were done on axial FLAIR images. Ideally, the measurement plane should not be limited to axial as the longest dimension can be in some other plane, and axial can underestimate the size. Any image (ax, sag or cor) that best represents the longest dimension should be used, and a perpendicular measurement done on the same image.

2. The size change for volumetric measurements uses a statistical change based on previously reported methodology rather than a standard percent cutoff. More clarification on this methodology is needed, as this could introduce discrepancies between assessments of different patients. Also, for volumetric measurements, the modified RANO criteria describes 40% increase or 65% decrease as cut-offs for progression and response respectively. Using those cut-offs would be a better way to compare to 25% increase of two dimensional measurements. (Neurotherapeutics. 2017 Apr; 14(2): 307–320.)

3. The clinical relevance of early identification/ smaller degrees of size change for LGG is yet to be established (as authors do acknowledge in the limitations), and it is unclear if early interventions for LGG based on volumetric measurements will lead to any outcome benefits. clinical criteria including seizures remain a strong contributor to overall response assessment for LGG. (Lancet Oncol 12:583-5932011) Please elaborate more on this.

4. Line 314 claims ' (delay in diagnosis by 2D methods) would result in a clinically significant delay in treatment'. This is unsubstantiated.

5. While many prior studies have had similar conclusions, about the greater reliability and reproducibility of volumetric over 2D measurements, the authors should elaborate on the practical barriers in implementation of volumetric techniques for routine and timely clinical decision making. The authors could also elaborate on some future directions, like deep-learning based segmentation, which could make it more accessible in future.

6. There is some repetition of 'results' in the 'discussion 'section, which should be avoided.

Author Response

Responses to reviewer #5

Reply to Point 1: We thank the reviewer for the comment on line 113. Although, we agree that measuring the lesion on the plane where the longest dimension is depicted, ideally from acquiring a 3D T2/FLAIR would have been optimal, our study is a retrospective study in which most of the cases had only axial 2D T2/FLAIR sequence available. The RANO definition for LGG tumor progression was used (25% increase of the product of the two-dimensional, perpendicular measurements of the lesion on T2/FLAIR sequence) consistently on the axial plane as it is stated on the manuscript. We have noted this in section 2.1, lines 135-136 in the manuscript.

Reply to Point 2: Thank you.  The modified RANO criteria described in Table 2 of Neurotherapeutics, 2017 Apr; 14(2): 307–320 introduce estimated volumetric change for CR (complete response), PR (partial response), PD (progressive disease) and SD (stable disease).  The criteria for the change in bidirectional product have not changed.  The PD 40% rule in 3D (1.118^3 = 1.3974) is an extension of the 25% rule in 2D (1.118^2 = 1.2499); they both represent an 11.8% increase in each dimension.  The latter, accepted by convention, lacks mathematical/statistical derivation and rigor.  Counterexamples arise from the fact that tumor growth is non-homogeneous; e.g. consider a tumor whose three measurements increase by 5%, 6%, and 25%.  The product 1.05*1.06*1.25 = 1.3913 is below the 40% 3D rule.  Because of the retrospective nature of our study and the lack of sagittal FLAIR sequences in all routinely acquired MRIs in current practice, we do not have the ability to evaluate the accuracy of the modified RANO criteria for estimated volumetric change.  The change-of-point method applies the same rigorous statistical standard to all patients and studies; instead of using the conventional RANO product rule in each dimension universally, the change-of-point statistical method determines if a current measurement is significantly different from all the measurements of the same patient (see section 2.1, lines 201-205 in text). 

Reply to Point 3:  We appreciate the insightful feedback provided by the reviewer. They have prompted us to elaborate on the importance of early detection of smaller degrees of size change for LGG, and we agree that while the clinical relevance of early detection of tumor growth has not yet been fully established, it is still a critical consideration for the development of new therapies (see section 4, lines 447 – 457 in text).

In fact, there is preliminary evidence to suggest that changes in tumor volume growth rates can be effectively used to evaluate the treatment response of LGG. Huang et al. conducted a study that demonstrated that untreated LGG shows a reduction in growth rates after following chemotherapy and/or radiotherapy.

As patients with LGG have long survival times, relying solely on overall survival (OS) and progression-free survival (PFS) as primary endpoints in clinical trials may not be optimal. Therefore, the proposed RANO response criteria for LGG, which are based on linear 2D measurements of T2/FLAIR signal abnormality, are not always accurate, especially due to the irregular shape of the tumors, post-treatment changes, interrater variability, and the presence of glial tissue. Instead, automated volumetric measurement of tumors can serve as a surrogate endpoint for treatment response/disease progression in clinical trials. This method allows for a more accurate assessment of changes in tumor size, which can be especially useful for evaluating the effects of novel therapies on LGG.

Another recent example is the ongoing INDIGO project, a phase III study of Vorasidenib (an oral, reversible, brain-penetrant inhibitor of mIDH1/2) in patients with LGG, in which volumetric MRI was used as a secondary endpoint, and as you pointed out, seizure activity was used as an exploratory endpoint (ref #30 in manuscript and Mellinghoff et al below). This highlights the growing recognition of the value of volumetric measurements in the development of novel therapies for LGG, particularly in the current rush to integrate AI 3D measuring tools into the PACS system.

A paragraph addressing this comment has been added to the discussion section of the manuscript. Additionally, the relevant suggested reference (Lancet Oncol 12:583-593, 2011) has also been added.

References

Huang, R.Y.; Young, R.J.; Ellingson, B.M.; Veeraraghavan, H.; Wang, W.; Tixier, F.; Um, H.; Nawaz, R.; Luks, T.; Kim, J.; et al. Volumetric analysis of IDH-mutant lower-grade glioma: a natural history study of tumor growth rates before and after treatment. Neuro Oncol 2020, 22, 1822-1830, doi:10.1093/neuonc/noaa105

Mellinghoff, I.K.; Bent, M.J.V.D.; Clarke, J.L.; Maher, E.A.; Peters, K.B.; Touat, M.; Groot, J.F.D.; Fuente, M.I.D.L.; Arrillaga-Romany, I.; Wick, W.; et al. INDIGO: A global, randomized, double-blind, phase III study of vorasidenib (VOR; AG-881) vs placebo in patients (pts) with residual or recurrent grade II glioma with an isocitrate dehydrogenase 1/2 (IDH1/2) mutation. Journal of Clinical Oncology 2020, 38, TPS2574-TPS2574, doi:10.1200/JCO.2020.38.15_suppl.TPS2574.

Reply to Point 4.  We thank the reviewer for bringing to our attention the lack of evidence to support the sentence on line 314. This issue will be addressed more thoroughly, as per point 3 outlined above (see section 4, lines 447 – 457 in text).  We have also modified the text as follows:

Compared to the volumetric ground-truth, the high false negative rate of up to 48% as well as a median time delay 1 to 2 years when RANO criteria detects growth after the volumetric assessment could result in a clinically significant delay in treatment, especially as new therapeutic agents are tested in LGG.

References

Huang, R.Y.; Young, R.J.; Ellingson, B.M.; Veeraraghavan, H.; Wang, W.; Tixier, F.; Um, H.; Nawaz, R.; Luks, T.; Kim, J.; et al. Volumetric analysis of IDH-mutant lower-grade glioma: a natural history study of tumor growth rates before and after treatment. Neuro Oncol 2020, 22, 1822-1830, doi:10.1093/neuonc/noaa105

Mellinghoff, I.K.; Bent, M.J.V.D.; Clarke, J.L.; Maher, E.A.; Peters, K.B.; Touat, M.; Groot, J.F.D.; Fuente, M.I.D.L.; Arrillaga-Romany, I.; Wick, W.; et al. INDIGO: A global, randomized, double-blind, phase III study of vorasidenib (VOR; AG-881) vs placebo in patients (pts) with residual or recurrent grade II glioma with an isocitrate dehydrogenase 1/2 (IDH1/2) mutation. Journal of Clinical Oncology 2020, 38, TPS2574-TPS2574, doi:10.1200/JCO.2020.38.15_suppl.TPS2574.

Reply to Point 5: We appreciate the point on future directions deep learning segmentation tools that could make volumetric analysis more accessible in the future. We have discussed several of these deep learning methods in section 4, lines 387 – 401 in the manuscript.

Reply to Point 6.  We have cleaned up the results that have been repeated in the discussion.

Once again, we would like to thank the reviewer for their feedback and insights, which have helped us to strengthen our manuscript.

Round 2

Reviewer 1 Report

Major requests were addressed by the authors.

Author Response

Thank you

Reviewer 5 Report

ok

Author Response

Thank you